# Simulated and Experimental Verification for a Terahertz Specific Finite Rate of Innovation Signal Processing Method

**DOI:** 10.3390/s22093387

**Published:** 2022-04-28

**Authors:** Xavier E. Ramirez Barker, Rayko I. Stanchev, Arturo I. Hernandez Serrano, Emma Pickwell-MacPherson

**Affiliations:** Department of Physics, University of Warwick, Coventry CV4 7AL, UK; x.barker@warwick.ac.uk (X.E.R.B.); rayko.stantchev@warwick.ac.uk (R.I.S.); arturo.hernandez-serrano@warwick.ac.uk (A.I.H.S.)

**Keywords:** finite rate of innovation, terahertz time-domain spectroscopy, low sampling rate, sum-of-sincs, annihilating filters

## Abstract

Recently, finite rate of innovation methods have been successfully applied to achieve low sampling rates in many areas, such as for ultrasound and radio signals. However, to the best of our knowledge, there are no journal publications applying this to real terahertz signals. In this work, we mathematically describe a finite rate of innovation method applied specifically to terahertz signals both experimentally and in simulation. To demonstrate our method, we applied it to randomized simulated signals with and without the presence of noise and to simple experimental measurements. We found excellent agreement between the simulated signals and those recreated based on results from our method, with this success also being replicated experimentally. These results were obtained at relatively low sampling rates, compared to standard methods, which is a key advantage to using a finite rate of innovation method as it allows for faster data acquisition and signal processing.

## 1. Introduction

Continuous signals must be reduced to a discrete form for experimental data acquisition and processing, a process called sampling the signal. Therefore, a sampling rate must be selected, balancing the ability to retain important features from the data set with the speed of data acquisition and size of the data set. A famous theorem on this aspect is the Shannon–Nyquist theorem, which states that for a perfect reconstruction of a bandlimited signal, a minimum sampling rate of double the maximal frequency is required. Thus, for secure retention of important data features, a floor is placed on the minimum sampling rate allowed, resulting in a constraint for data acquisition speed.

Recently, finite rate of innovation (FRI) [1] theory has been employed to achieve low sampling rates, with respect to the relevant traditional sampling schemes, in both ultrasound simulated and experimental data [2], as well as in a simulated terahertz (THz) context in communications application [3]. It has also been shown to handle the reconstruction of sparse signals robustly, such as in [4] with the application for source resolution in radioastronomy showcased. Furthermore, in a frequency-domain optical coherence tomography context, an FRI method has achieved improved resolution and signal-to-reconstruction noise compared to the standard approach [5]. This theory utilizes signals with a finite number of degrees of freedom per unit time, which pulse signals, such as those used in ultrasound and THz time-domain spectroscopy. This concept will be further explained and mathematically defined in the following section. Given knowledge of the pulse shape and number of reflections or pulses expected, a sampling rate below the Shannon–Nyquist limit can even be employed to achieve a full reconstruction of the sampled data [3].

In this paper, FRI will be applied to both simulated and basic experimental THz data. THz covers the electromagnetic spectrum from frequencies of around 0.1 THz to 10 THz and has many inherent advantages by being non-ionizing, non-destructive and providing noncontact modalities. Therefore, THz has had great research interest in areas such as biomedical physics, security applications and material characterization over the past years [6]. One constraint especially significant for biomedical in vivo imaging is the duration of the scan. For example, in the protocol our group developed for assessing skin hydration [7], the patient was required to hold their arm stationary during the measurements, with any movement possibly affecting the recorded data. Thus, with a lower sampling rate these scans could be taken more quickly with less risk of this occurring and with the additional benefit of any time-dependent effects, such as occlusion of the skin [8], being minimized.

In typical terahertz time-domain spectroscopy (THz-TDS) setups, a femtosecond laser is split into a pump and a probe beam. The pump beam excites the photoconductive emitter whilst the probe beam is controlled by an optical delay line to excite the detector at different THz signal time steps [9]. Therefore, in the case of THz-TDS, the steps along the time delay line constitute the sampling points that the FRI method shown in this paper will seek to minimize. In addition to the standard measurement systems, this method could see use in single-pixel spectroscopic imaging methods [10] by allowing for a lower sampling rate of temporal waveforms per pixel.

Our aim is to achieve lower sampling rates compared to current popular methods by utilizing FRI theory whilst maintaining similar experimental methodology and required foreknowledge of the sample. Although FRI has previously been applied to other fields such as ultrasound and to simulated THz data in communication applications, this is the first journal publication applying it to experimentally obtained THz-TDS data. By employing our THz FRI based method, we achieved low sampling rates in both simulated and experimental THz data sets whilst demonstrating agreement with the original simulated and measured signals, respectively.

## 2. Theory

The first building block for creating this FRI method is to simulate the THz pulse shape we expect from our experiment in a form which is compatible with later mathematical manipulations. In the frequency domain, sum-of-sincs (SoS) is the usual form utilized as the sampling kernel for these methods [11]. This is because the kernel approximates a reflection or transmission response, and therefore is able to solve both non-periodic and periodic cases whilst having a finite duration itself which is easily mathematically manipulated. It is defined in general terms by [2]:(1)H(ω)=∑p∈Πdpsinc(ω2πτ−p),
where p is an integer in the chosen set of integers Π, ω is the frequency and τ is the period containing an entire repetition of the SoS as this forms the repeating sampling kernel required for this method. The chosen integer number sets Π and $d_{p}$ are free parameters optimized for the specific application explored later in this section. In this work, we applied this method mainly in the time domain, as the experimental data used in this work as well as all THz-TDS are acquired in that domain. Thus, we required the time domain version of (1):(2)$$h\left(t\right)=\mathrm{rect}\left(\frac{t}{\tau }\right)\underset{p\in \Pi }{\sum }d_{p}e^{i\frac{2\pi pt}{\tau }},$$
where t is the time and $\mathrm{rect}\left(\frac{t}{\tau }\right)=1$ for $-\frac{\tau }{2}\leq t\leq \frac{\tau }{2}$ whilst being zero elsewhere, limiting the time range to only containing one repetition of the sampling kernel. The result of setting the free parameters $\left\{d_{p}\right\}$ to 1 and choosing Π={−P,…,P} where P=25 can be seen in Figure 1a. Here, the central peak is surrounded by side lobes being far from the real THz pulse representation, demonstrating the need for further work to mold our sampling kernel. This comes in the form of applying a length-N symmetric Hamming window for the free parameters:(3)dp=0.54−0.46cos(2πp+N2N),        p∈Π, where the cardinality N=|Π|. Figure 1b shows the result of applying this Hamming window, with the side lobes being smoothed out to give a closer appearance of a single peak. To generate a sampling kernel which represents a single THz pulse, two of these SoS peaks were combined by offsetting and scaling one of the SoS pulses, as shown in Figure 1c. The offset between the pulses was equal to the width of the center peak at 0 amplitude and the relative scaling difference was a factor of −0.6. These were selected by comparison to a real THz reference and can be automatically tailored to a specific experimental reference as demonstrated later in this paper.

Finite streams of pulses can be used to represent THz-TDS data, finite by virtue of the data acquisition range used, and the stream of pulses representing the THz waveforms constituting the data, i.e., reflections off boundaries between materials of different refractive indices. Therefore, let us consider a τ-periodic stream of L pulses with amplitudes al located at distinct times tl:(4)x(t)=∑m∈ℤ∑l=1Lalh(t−tl−mτ).

Here, h(t) is a known pulse shape, which in our case has been constructed and defined in the previous paragraphs and is in the form of (2). For this consideration, we also have the constraints of tl ∈ [0,τ), al ∈ ℂ, l=1 . . . L, {tl,al}l=1L and an additional constraint on N≥|Π|≥2L. Given that we have L pulses which are each fully described by two parameters, the amplitude and time location, we have 2L degrees of freedom per unit time and so a finite rate of innovation, ρ, of:(5)ρ=2Lτ.

With the aim of achieving the minimum sampling rate whilst being able to adequately reconstruct the sample, we can target 2L samples per τ. This is the ideal minimal number of samples required using this method, which can result in sub-Nyquist rates. However, this ideal sampling rate does not account for issues such as sampling points not containing useful information, i.e., not lying on the pulse, or the presence of noise and other aberrations. By defining the periodic extension of our pulse shape h(t) as:(6)g(t)=∑m∈ℤh(t−mt).

We can apply Poisson’s summation formula [12] to rewrite (6) above as:(7)g(t)=1τ∑k∈ℤH(2πkτ)ei2πktτ,
where H(ω) represents the Fourier transform of h(t). By substituting this result into (4):
(8)x(t)=∑l=1Lalg(t−tl)=∑k∈ℤ(1τH(2πkτ)∑l=1Lale−i2πktlτ)ei2πktτ=∑k∈ℤX[k]ei2πktτ, where we have used X[k] to denote the bracketed terms in the preceding line, which are the Fourier coefficients of that line, it can be shown [1] that once at least 2L Fourier coefficients are known, the amplitudes and time locations of the stream of pulses representing our data can be found. This enables the reconstruction, or estimation, of our data, given a suitable sampling kernel.

We now require a way to calculate what these Fourier coefficients are. We begin by considering the uniform sampling of signal x(t) of the form seen in (4) with a sampling kernel in the form of (2), which gives a sufficient characterization of x(t) with uniform samples N at locations t=nT:(9)$$y_{n}=\langle h\left(t-nT\right),x\left(t\right)\rangle =\int_{-\infty }^{\infty }h\left(t-nT\right)x\left(t\right)dt,\;\;\;\;\;\;\;\;\;n=0,\ldots ,N-1.$$

By substituting (8) into (9):(10)yn=∑k∈ℤX[k]⟨h(t−nT),ei2πktτ⟩=∑k∈ℤX[k]H(2πkτ)ei2πknTτ=∑k=−LLX[k]ei2πknTτ.

Here, when T is a divisor of τ, as it is in our case, this reduces line 2 in the above equation to the inverse discrete-time Fourier transform of X[k] resulting in line 3 [1]. Utilizing Prony’s method [13], we now introduce the annihilation filter, A[k], stage of the method [14], which is by definition required to satisfy the convolution:(11)A[k]∗X[k]=0.

By satisfying this constraint, and in the case of our chosen sampling kernel with X[k] reducing to:(12)X[k]=1τ∑l=1Lale−i2πktlτ,        k∈ℤ.

We can present A[k] in the form of its z-transform:(13)$$A\left[z\right]=\overset{L}{\underset{k=0}{\sum }}A\left[k\right]z^{-k}.$$

This is because A[z] has L zero valued null terms at cl=e−i(2πtlτ), allowing A[k] to be represented by the convolution of L elementary filters [15], each of which zero out one of the sum of L exponentials in X[k] for the convolution in (11). Now, we can construct a rectangular Toeplitz matrix, X, from X[k]:(14)$$X=\left[\begin{array}{ccccc} X\left[-M+K\right] & X\left[-M+K-1\right] & \cdots & \cdots & X\left[-M\right]\\ X\left[-M+K+1\right] & X\left[-M+K\right] & \cdots & \cdots & X\left[-M+1\right]\\ \vdots & \ddots & \ddots & \ddots & \vdots \\ \vdots & \ddots & \ddots & \ddots & \vdots \\ X\left[M\right] & X\left[M-1\right] & \cdots & \cdots & X\left[M-K\right] \end{array}\right],$$
where M,K≥L. Note that the matrix has a number of columns equal to K+1 and a number of rows equal to 2M−K+1. Along with the matrix form of A=[A[0],A[1],⋯,A[L]] we can solve equation (11), with the additional constraint for our sampling number of N≥2M+1, by performing the singular value decomposition [16] of X and selecting the eigenvalues corresponding to the smallest eigenvector, giving the annihilation filter coefficients for A. This allows us to find the roots $c_{l}$ of A[z], with obtaining the time locations $t_{l}$ clearly following. With the time locations found, the last piece of information needed is the amplitudes, $a_{l},$ which can be calculated using the Vandermonde system [17]:(15)$$\left[\begin{array}{c} X\left[0\right]\\ X\left[1\right]\\ \vdots \\ X\left[L-1\right] \end{array}\right]=\frac{1}{\tau }\left[\begin{array}{cccc} c_{0}^{0} & c_{1}^{0} & \cdots & c_{L-1}^{0}\\ c_{0}^{1} & c_{1}^{1} & \cdots & c_{L-1}^{1}\\ \vdots & \vdots & \cdots & \vdots \\ c_{0}^{L-1} & c_{1}^{L-1} & \cdots & c_{L-1}^{L-1} \end{array}\right]\;.\;\left[\begin{array}{c} a_{0}\\ a_{1}\\ \vdots \\ a_{L-1} \end{array}\right],$$
where the exponent denotes the power to which the term is taken to. As we have distinct $t_{l},$ this system always has a solution, providing our amplitudes.

## 3. Simulation Results

### 3.1. Noiseless THz Model

To verify this FRI method for THz signals, a simulated signal was generated using the same method as the previously described THz appropriate sampling kernel creation. Five Dirac peaks were randomly generated, each representing the amplitudes, $a_{l}$, and time positions, $t_{l}$, of simulated THz pulses, so L=5. τ was arbitrarily chosen to be 1, resulting in an FRI of 10. M and P were both taken to be 5L and K taken as L; these values were selected according to the limitations outlined for them in the previous section and by balancing the code runtime with the quality of our results. Larger values for M, P and K resulted in longer runtimes but provided better-quality results. These Diracs representing the amplitudes and time locations are shown by the blue peaks in Figure 2a, with the resulting simulated THz data shown in Figure 2b by green. This was then uniformly sampled as seen in the 25 red data points in Figure 2b. By following the methodology outlined in the previous section, we obtained the orange Dirac peaks in Figure 2a, which represent the estimated amplitude and time location of the THz pulses that constituted our simulated signal. As we know the sampling kernel, we can recreate the original simulated THz data which is shown by the dashed blue line in Figure 2b.

In both representations of the output data, it can be seen that there is close agreement between the estimated output and the simulated input; in Figure 2a the amplitudes and time locations of the Dirac peaks closely match, and this is further demonstrated by comparing the reconstructed signal with the original in Figure 2b. As the pulse shape used to create the simulated data is exactly known, the difference between the reconstructed and original signal is an effective measure of the performance of our method. By contrasting the last pulse in Figure 2b, which is completely isolated, with the two pairs of closely neighboring pulses the rest of the data consists of, it is demonstrated that, even when the pulses are overlapping and not distinguishable by eye, the amplitudes and time locations can still be found with great accuracy. Crucially, this has all been achieved whilst using a relatively low sampling rate of a total of 25 points, with it been further seen that there are only a few sampling points describing each of the pulses. In particular, there are only three or four sampling points that fall within the simulated pulses; however, we are still able to extract the pulses’ exact amplitudes and time locations. This exemplifies a key benefit of this method, low sampling rates, which allow for faster data acquisition without losing information about our signal.

### 3.2. Simulated Noise THz Model

No experimental data are completely free of noise, so to simulate this effect Gaussian white noise was added to result in a signal to noise ratio of 6 dB, the result of which can be observed on the sampling points in Figure 3b. Although this is a significant amount of noise, more so than would usually accompany experimental THz measurements such as those shown later in this paper, the reconstruction is shown to almost negate the effect of the noise completely. This is demonstrated by comparing the processed Dirac results, shown in Figure 3a, to those in the noiseless model seen in Figure 2a. We can see that in both models the processed Diracs are close to the original Dirac peaks whilst also being very similar to each other. Additionally, by observing how closely the reconstructed and the original signals match in Figure 3b, further evidence for the effectiveness of removing the effect of noise is given.

## 4. Experimental Results

We have shown the great potential of our FRI model in the previous section, by achieving an accurate reconstruction of the simulated signal both without and with the presence of white Gaussian noise whilst using a relatively low sampling rate. However, there are further challenges to applying this method to experimental results; primarily, the accuracy of our sampling kernel in representing the THz pulse and accounting for how this pulse would change shape during transmission and reflection through different materials.

A TeraPulse 4000 from TeraView Ltd. was used in reflection geometry with an incidence angle of 30° to measure the air-plastic reflection off a thick piece of plastic. By using this as an experimental reference, the method described in Section 2 was employed to create a SoS which closely resembled the shape of this reference. Ideally, the SoS sampling kernel created in this way would closely resemble the pulse in our simple experimental data obtained from a thick plastic sheet. This was measured using the same experimental procedure used for the reference acquisition and with the interface causing reflections between both air and plastic. The SoS form of our reference is required in most of the FRI method, except for the final stage of calculating the amplitudes. Instead of using the Vandermonde system, a standard least-squares minimization technique between the original measured signal and measured reference, repeated at the time locations found in the previous step, was used. For the recreation of the signal, we used the time location and amplitude estimates along with the measured reference. This achieved a signal shape more similar to the original, as the measured reference more closely resembled the correct shape.

Figure 4a contains this simple experimental data. The raw measurement is shown by the green line, with the reduced sampling points used as the input for our method shown by the red points and the reconstructed output of the FRI method shown by the blue dashed line. It can be seen that there are relatively few data points describing each of the two reflection pulses, but despite this the reconstruction is a close match to the original experimental data. This indicates that the time locations and amplitudes of the reflections found by our method are accurate, as by using these in combination with our measured reference as the basis for the reconstruction we obtained a similar result to the original data. However, it can be noticed that there is a mismatch in the amplitudes of the second halves of the pulses between the raw experimental data and the reconstructed result from our model. As the amplitudes for the first halves of the pulses are a close match, this is likely the result of an imperfect estimation of the pulse shape. Compared to the standard amount of data points the THz system we were using measures, we down-sampled by a factor of over 45 to obtain the sampling points shown, which resulted in a time sampling interval of 0.33 ps. This indicates that by using this method, fewer data points can be measured and thus a much shorter data acquisition time can be achieved.

Figure 4b shows the frequency domain version of the data presented in Figure 4a, after it has undergone a fast Fourier transform (FFT) and been normalized. The low sampling rate FFT, shown by the red line, begins to diverge from the fully sampled experimental data FFT, shown in green, from around 0.7 THz. For frequencies larger than 1.3 THz there is mostly a very significant difference in the normalized FFT amplitudes, showing the frequency domain inaccuracy of the low sampling rate data at these frequencies. This is because the sampling interval of 0.33 ps corresponds to a Nyquist sampling frequency of 1.51 THz. Crucially, the FFT of the reconstruction from the low sampling points using our FRI method, shown by the blue line, does not share this divergence and inaccuracy. This demonstrates that our method accurately reconstructs the frequency domain data of THz-TDS measurements taken at sampling rates below the Nyquist frequency, providing the benefit of quicker measurements whilst ensuring the retention of frequency domain information.

More complicated experimental situations were investigated; however, it was found that for low sampling rates the SoS sampling kernel proved to be too rigid to account for the varying pulse shape as it reflected off boundaries and propagated through different materials. Thus, an interesting avenue of future research would be to investigate more accommodating sampling kernel models for THz pulses. Despite this issue, we have shown the great potential of this method to achieve accurate experimental data processing at low sampling rates, allowing for quicker data acquisition and processing once a more versatile sampling kernel has been developed.

## 5. Conclusions

By demonstrating accurate results with relatively low sampling rates in both simulated and simple experimental data sets, the FRI method described in this paper has been verified and the potential for its application in more complicated THz experiments has been shown. Furthermore, looking at the FFT spectra of the low-sampled experimental data, we see that information above and around the Nyquist frequency was lost whereas the FRI reconstruction retains the frequency information of the original data. The potential of this method primarily lies with the low sampling rates unlocked, which are especially attractive in applications where the sample has a time dependency or other applications where quicker data acquisition would be of large benefit. For instance, in our research group’s in vivo experiments for human skin, patients are required to remain very still during the data acquisition. The shorter this period, the less likely a patient is to accidentally shift, causing an effect on the measured data. However, when more complicated experimental data sets were investigated, we discovered that further work is necessary to create a versatile sampling kernel which accounts for the THz pulse shape changing during propagation through dispersive samples and reflection off different material boundaries. As the method has performed very strongly in simulation and simple experimentation, we believe that further research into creating this sampling kernel method is of great interest, bringing the benefits of fast data acquisition and processing to more complicated and interesting experimental situations.

## Figures and Tables

**Figure 1 sensors-22-03387-f001:**
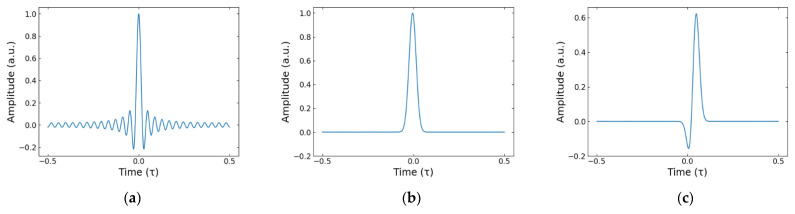
The progression of the sampling kernel through our methodology. (**a**) Time domain response for SoS sampling kernel. Described by h(t) from (2) for P=25. (**b**) The resulting kernel after a Hamming window is applied to the SoS sampling kernel shown to the left. (**c**) Our THz-like sampling kernel. Created by combining two Hamming windowed SoS sampling kernels.

**Figure 2 sensors-22-03387-f002:**
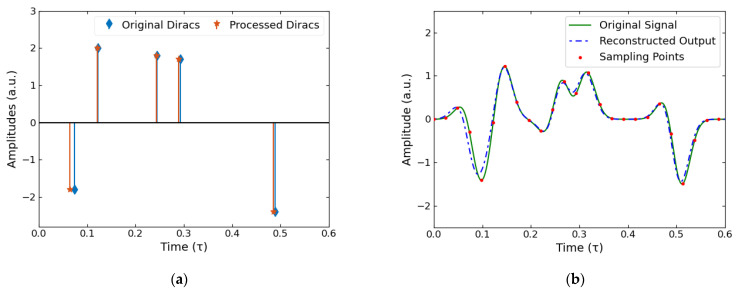
Noiseless simulated results. (**a**) The blue peaks represent the simulated THz pulses’ time locations and amplitudes with the orange Diracs being the FRI method processed results. (**b**) The simulated signal (green line) was sampled (red points) to input into the FRI code. The reconstructed signal (dashed blue line) was calculated using the calculated time locations and amplitudes shown in (**a**) along with the known sampling kernel.

**Figure 3 sensors-22-03387-f003:**
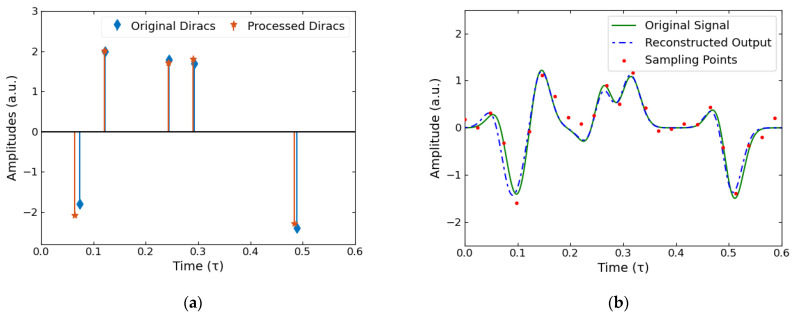
Simulated results with a Gaussian white noise model to give a signal to noise ratio of 6 dB. (**a**) Simulated results in the presence of noise (orange), with the original (blue) for comparison. (**b**) Reconstruction of the simulated signal with the presence of noise (dashed blue line), compared to the original noiseless signal (green line) and the sampling points (red points).

**Figure 4 sensors-22-03387-f004:**
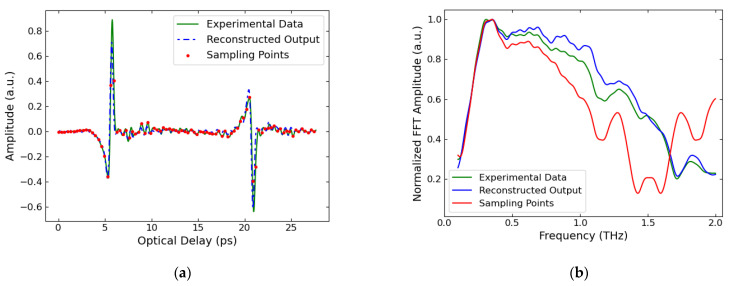
Experimental verification of our THz FRI method. (**a**) The raw data, shown by the green line, are reflections off the sides of a thick plastic block. The sampling points are shown by the red dots, which were used as the input for the method. The resulting reconstruction is shown by the dashed blue line. (**b**) The normalized FFT spectra for these data.

## Data Availability

The data presented in this paper is available at: https://doi.org/10.6084/m9.figshare.19204992 (accessed on 8 March 2022).

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
