# Peer review of "Simulated and Experimental Verification for a Terahertz Specific Finite Rate of Innovation Signal Processing Method"

_sensors, 2022, doi:10.3390/s22093387_

Round 1
Reviewer 1 Report
Reference: sensors-1652227
Review Report
The MS entitled “Simulated and Experimental Verification for a Terahertz 2 Specific Finite Rate of Innovation Signal Processing Method” by X.E. Ramirez Barker and co-workers reports on the use of a new method, finite rate of innovation to the analysis of terahertz (THz) signals.
The paper is well written. Both the model, the procedure and the obtained results are well sound. The method has been applied first to a noiseless terahertz theoretical model and subsequently to a model that included Gaussian with noise with reasonably good results. Finally, the new method was applied to experimental data from spectroscopic measurements carried out on a plastic sample using a THz-TDS (Time Domain Spectroscopy).
The results are promising but, in my opinion, authors should provide the Fourier transforms of Figures 2 (b), 3 (b) and 4 in order to really appreciate the impact in the frequency domain of the differences seen in the time-domain between the reconstructed signals (using the new method) and the conventional (both theoretical and experimental) ones.
Author Response
Thank you for this useful suggestion. We have extended our analysis to the frequency domain for our experimental data presented in this manuscript. This is shown in the edited manuscript, with the addition of Figure 4 (b) containing the normalized amplitudes for the FFT of all the data presented in Figure 4 (a). It can be seen that the FFT of the reconstruction using our FRI method contains much more accurate frequency information than the low sampling of the data itself. The sampling rate previously used for this data was further reduced to better demonstrate the results of our method at low sampling rates. For the simulated data the results were less interesting and not included in the edited manuscript. As the simulated pulse was created to resemble THz-TDS data in time domain only, the frequency profile does not resemble that of THz data. The FFT of the simulated data, low sampling of the simulated data used in our method and the reconstruction after the method almost exactly match.
Reviewer 2 Report
The manuscript , entitled 'Simulated and Experimental Verification for a Terahertz specific finite rate of innovation signal processing method, describes the applicability of finite rate of innovation method in the THz signals. The idea of low sampling rates is successfully applied in many areas of reserach, such as ultrasound and radio signals. The authors propose a complete study of simulated and experimental verification for THz signal. The work is well-written, and well-discussed. This work is highly relevant and of great interest to the THz community, and it is appropriate for publication.
Author Response
Thank you for your positive feedback and comments.
Reviewer 3 Report
The authors mathematically describe a finite rate of innovation (FRI) method, that can be used to achieve low sampling rates in many areas, applied specifically to terahertz (THz) signals. By demonstrating accurate results with relatively low sampling rates in both simulated and simple experimental data sets, the FRI method described in this manuscript has been verified and the potential for its application in more complicated THz experiments has been shown.
The low sampling rates the method unlocks are attractive in applications where the sample has a time dependency or other applications where quicker data acquisition would be of large benefit.
Overall, the topic of the manuscript is interesting and the result sheds new light on terahertz science and technology community. The formula used in the manuscript is reliable. I think the results are sound and deserves publication.
Author Response
Thank you very much for your positive comments and feedback.
Reviewer 4 Report
In this manuscript, the authors use finite rate of innovation methods in the reconstruction of THz time domain signals. The use of this methodology would improve the acquisition time for THz signal, which is of great interest for those who work in the area, especially for image acquisition or faster measurements of specific biological applications. The presented results show potential, although further research must be done to present a robust methodology for all applications, as the authors state in the work.
I suggest for the manuscript to be considered for publication after the following questions and recommendations are addressed by the authors:
- The presented results were obtained for a sampling rate of 25 points. I understand that a low sampling rate is desired given the goal of the method and, if I understood correctly, the difference with a higher sampling rate would be a longer code runtime. However, I think it may be useful to understand how a lower/higher sampling rate would affect the outcome of the calculations. This would be useful to the reader or the user of the method, given that, for different applications sampling rate could be increased if higher quality is needed in the results or viceversa. For this, the authors can include in their analysis the results for different sampling rates, and a comparison of quality of results and runtime in each case.
- I would also like to suggest for the authors to extend their analysis to the effects on the frequency domain of the data. There are some differences at the amplitude obtained in the reconstructed output for the experimental case and this would impact further calculations for sample characterization using frequency data.
Author Response
Thank you for your helpful suggestions - they are in line with those of Reviewer 1. We have extended our analysis to the frequency domain for our experimental data presented in this manuscript. This is shown in the edited manuscript, with the addition of Figure 4 (b) containing the normalized amplitudes for the FFT of all the data presented in Figure 4 (a). It can be seen that the FFT of the reconstruction using our FRI method contains much more accurate frequency information than the low sampling of the data itself. The sampling rate previously used for this data was further reduced to better demonstrate the results of our method at low sampling rates.
We investigated the effects of using higher sampling rates. We found that for the simulated data presented in our manuscript there was no significant improvement to the amplitudes and time locations of the reflections determined when higher sampling rates were used. We will further investigate this in the future, to modify our method to achieve better performances at higher sampling rates.
Round 2
Reviewer 1 Report
I thanks the author for their effort and staisfactory answer to my questions.